# Prevalence and inequality in persistent undiagnosed, untreated, and uncontrolled hypertension: Evidence from a cohort of older Mexicans

C. M. Dieteren[1]*, O. O'Donnell[1,2,3], I. Bonfrer[1]

1 Erasmus School of Health Policy & Management, Erasmus University Rotterdam, Rotterdam, the Netherlands, 2 Erasmus School of Economics, Erasmus University Rotterdam, Rotterdam, the Netherlands, 3 Faculty of Business and Economics, University of Lausanne, Lausanne, Switzerland

* dieteren@eshpm.eur.nl

**Data Availability Statement:** The WHO SAGE data is publicly available via the following link: https://

## Abstract

Hypertension is the leading risk factor for cardiovascular diseases (CVDs) and substantial gaps in diagnosis, treatment and control signal failure to avert premature deaths. Our aim was to estimate the prevalence and assess the socioeconomic distribution of hypertension that remained undiagnosed, untreated, and uncontrolled for at least five years among older Mexicans and to estimate rates of transition from those states to diagnosis, treatment and control. We used data from a cohort of Mexicans aged 50+ in two waves of the WHO Study on Global AGEing and adult health (SAGE) collected in 2009 and 2014. Blood pressure was measured, hypertension diagnosis and treatment self-reported. We estimated prevalence and transition rates over five years and calculated concentration indices to identify socioeconomic inequalities using a wealth index. Using probit models, we identify characteristics of those facing the greatest barriers in receiving hypertension care. More than 60 percent of individuals with full item response (N = 945) were classified as hypertensive. Over one third of those undiagnosed continued to be in that state five years later. More than two fifths of those initially untreated remained so, and over three fifths of those initially uncontrolled failed to achieve continued blood pressure control. While being classified as hypertensive was more concentrated among the rich, missing diagnosis, treatment and control were more prevalent among the poor. Men, singles, rural dwellers, uninsured, and those with overweight were more likely to have persistent undiagnosed, untreated, and uncontrolled hypertension. There is room for improvement in both hypertension diagnosis and treatment in Mexico. Clinical and public health attention is required, even for those who initially had their hypertension controlled. To ensure more equitable hypertension care and effectively prevent premature deaths, increased diagnosis and long-term treatment efforts should especially be directed towards men, singles, uninsured, and those with overweight.

apps.who.int/healthinfo/systems/surveydata/index.php/catalog/sage.

**Funding:** O'Donnell was partly funded by the Swiss Agency for Development and Cooperation / National Science Foundation Programme for Research on Global Issues for Development, grant 400640_160374. Igna Bonfrer acknowledges funding from the Research Excellence Initiative on Universal Health Coverage from the Erasmus University Rotterdam, the Netherlands. The funders had no role in study design, data collection and analysis, decision to publish, or preparation of the manuscript.

**Competing interests:** The authors have no conflict of interests to declare.

## Introduction

Hypertension is the leading global risk factor for cardiovascular diseases (CVDs) [1]. Worldwide, there are substantial gaps in diagnosis, treatment, and control of hypertension [2–7], signaling failures to prevent CVDs and avert millions of premature deaths [8]. In middle-income countries, where hypertension prevalence is rising [9, 10], populations are ageing, and health systems are straining to cope with the double burden of disease, gaps in diagnosis and management of hypertension [5, 11–13] can take a heavy toll on population health.

In high-income countries, hypertension tends to be more prevalent among lower socioeconomic groups [14]. In low- and middle-income countries (LMIC), evidence on the socioeconomic gradient in hypertension is mixed, which may reflect changes in the gradient as countries move through the epidemiological transition [2, 15–17]. There is evidence, however, that the socially disadvantaged in LMIC have worse access to hypertension care [5, 18] and so potentially suffer great ill-health as a consequence of uncontrolled hypertension. More effective and equitable targeting of hypertension screening and treatment requires improved understanding of the sociodemographic groups that face the greatest barriers in accessing these services.

In Mexico, which has the highest prevalence of overweight in the world [19] and non-communicable diseases (NCDs) account for 80% of all deaths [20], estimated hypertension prevalence in the adult population aged 18 years and older was 25.5% in 2016, and increased substantially with age with a prevalence near 50% at the age of 60 [21, 22]. Among adults with hypertension, 40% were estimated to be undiagnosed, 21% were untreated, and 55% had not achieved blood pressure control in 2016 [21].

Estimates of diagnosis, treatment, and control of hypertension are valuable for monitoring and targeting of CVD prevention. However, the cross-sectional nature of most of this evidence is limiting in two respects. First, it does not provide information about persistent undiagnosed, untreated, and uncontrolled hypertension. Given that the risks of severe health consequences rise steeply with the duration of exposure to uncontrolled hypertension [23], it is important to establish prevalence of the condition that remains undiagnosed, untreated, and uncontrolled for an extended period of time and how these prevalence rates vary with sociodemographic characteristics. Second, a cross-sectional approach does not allow estimation of rates of transition from undiagnosed to diagnosed, untreated to treated, and uncontrolled to controlled hypertension, nor can it reveal reverse transitions from treated to untreated and controlled to uncontrolled, both of which indicate failures in hypertension management and treatment adherence.

These limitations can be addressed by following a cohort over time to identify the proportion and type of people who remain undiagnosed, untreated, and uncontrolled for an extended period, as well as rates of transition to more favorable states. This study aimed to estimate the prevalence and socioeconomic distribution of hypertension that remained undiagnosed, untreated, and uncontrolled for at least five years among Mexicans aged 50 years and older and to estimate rates of transition from those states to diagnosis, treatment, and control. To help target improvements in hypertension screening and management on vulnerable groups, we aimed to identify sociodemographic characteristics associated with remaining in an unfavorable hypertension state.

## Methods

### Sample

We used longitudinal data from the Mexican sample of the World Health Organization (WHO) Study on Global AGEing and adult health (SAGE) [6]. Our study focused on adults

aged 50 years and older. Mexico is the only one of six countries participating in SAGE to have made longitudinal data publicly available (as of 2021).

The sample for Wave 1 (November 2009–January 2010) was based on the 2003 WHO World Health Survey (WHS) for Mexico (hereafter, Wave 0). A total of 96 strata were defined over 32 states and three levels of urbanicity (rural, urban, and metropolitan) (6). A nationally representative sample was obtained in Wave 0 by conducting cluster random sampling with Basic Geo-Statistical Areas forming the primary sample units (PSUs). In total, 40,000 households were randomly sampled [24]. To obtain the Wave 1 sample, probability sampling was used to select 211 PSUs from the 797 sampled in Wave 0 [25]. In each selected rural and urban PSU, all Wave 0 individuals who had been aged 50 years or older in 2003 were included in the Wave 1 target sample (ibid). In each of the selected metropolitan PSUs, a random sample of 90% of individuals aged 50 years and older in 2003 were included. In addition, a systematic sample of 1000 individuals from Wave 0 who had been aged 18–49 across all selected PSUs were included as the primary sample.

Wave 1 had a relatively low response rate of 53%. Response was lower for middle-aged adults aged 50–59 years (42%) than for younger adults aged 18–49 years (58%). The low response rate has been attributed to the short time available for field work, which left little time to revisit sampled households that did answer during the initial visit in this wave [6], but no further information on the average characteristics of those missing has been made available. An interval of six-seven years between Wave 0 and Wave 1 also contributed to a high rate of attrition and a low response in the latter wave [6].

SAGE Wave 2 was conducted in July-October 2014. The target sample included all individuals who participated in Wave 1, plus those aged 50 years and older in 2014 who were not in the Wave 0 (or Wave 1) sample but who lived in a household that included someone from that sample [6]. The Wave 2 response rate was 83% for households and 81% for individuals.

The SAGE sample was designed, after weighting, to be nationally representative for the population aged 50 years and older at the time of each wave. We restricted the analysis sample to this age range and to respondents observed in both Wave 1 and Wave 2. To maximize the size of this cohort, we selected respondents aged 50 years and older in Wave 2 who also participated in Wave 1. Some of these respondents were therefore slightly younger than 50 in Wave 1. Then, we excluded respondents that had missing data on any of the hypertension measurements. The final step was to exclude respondents that had missing data on any of the other relevant covariates in Wave 2 (see Flow Chart Fig 1).

## Measurements

**Hypertension.** Blood pressure (BP) was measured using a Boso Medistar Wrist Blood Pressure Monitor Model S during a home visit [26]. Three measurements were taken, with a minimum of one minute between each. Each participant was asked: "Have you ever been diagnosed with high blood pressure (hypertension)?" A positive response was followed with: "Have you been taking any medications or other treatment for it during a) the last 2 weeks, b) the last 12 months?" We classified a participant as having hypertension (HTN) if a) the last two measurements gave a mean systolic BP $\geq$140 mm Hg or mean diastolic BP $\geq$ 90 mm Hg, or b) they reported ever having been diagnosed with high BP [13, 27]. Those classified as having hypertension were then categorized as: a) *diagnosed*, if they reported ever having been diagnosed (*HTN Diagnosed*); b) *treated*, if they reported taking medication or another treatment (*HTN Treated*); and, c) *controlled*, if they had measured systolic BP < 140 mm Hg and measured diastolic BP < 90 mm Hg (*HTN Controlled*). The other respondents, either with or

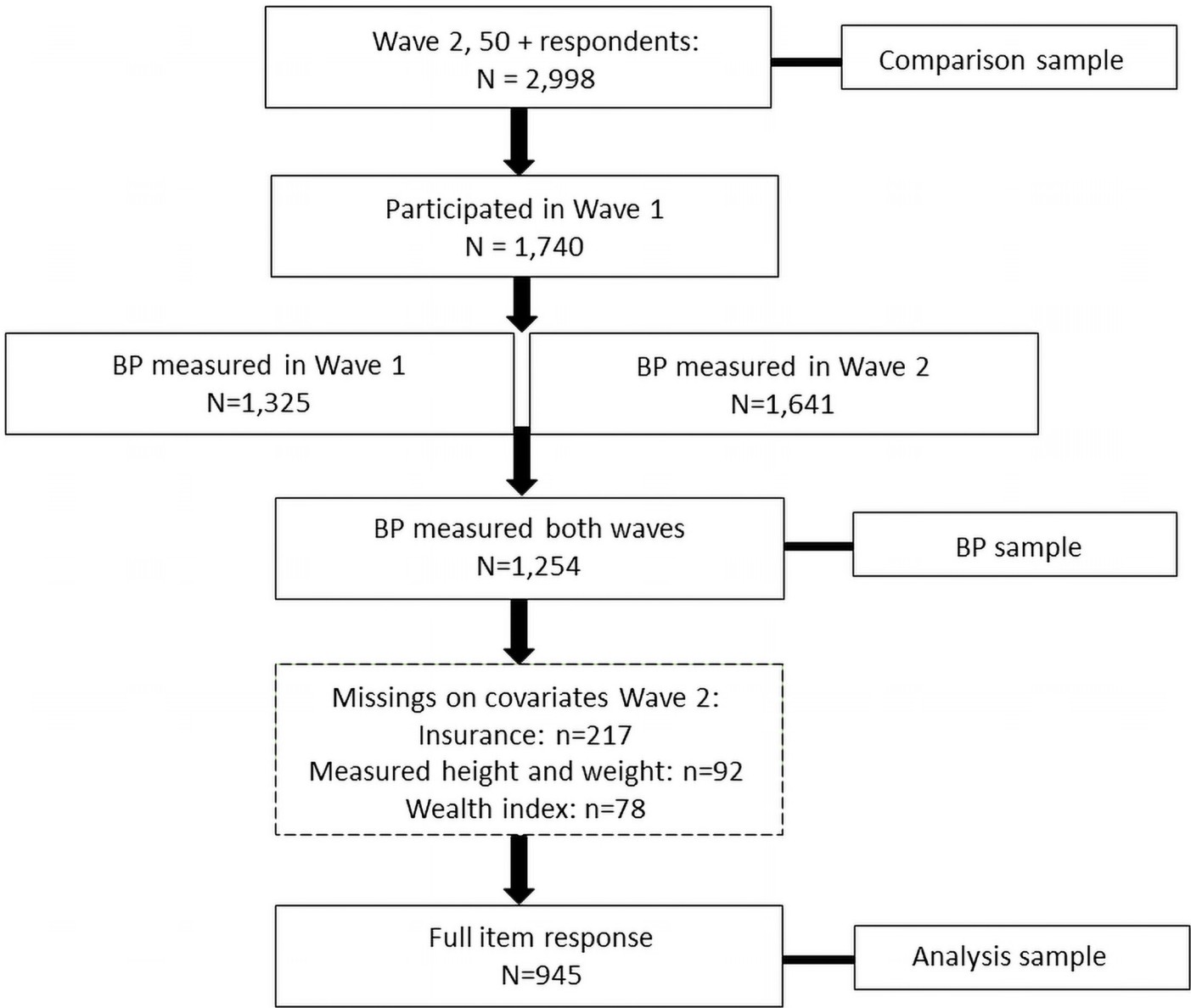

**Fig 1. Participant flow chart.** *Notes*: BP measured indicates that BP was measured and hypertension diagnosis and treatment were reported, allowing hypertension status to be established.

without a classification of hypertension, were classified as being undiagnosed, untreated, or uncontrolled defined analogously (see Table 1).

While we recognized that clinically diagnosed hypertension is a chronic condition, we did not classify a participant as necessarily having hypertension in Wave 2 if they were classified with the condition in Wave 1. The reason was that we did not observe clinical diagnoses made on the basis of BP measurements on multiple occasions. Measured BP $\geq$140/90 mm Hg on a single occasion could be a false positive. By classifying respondents in each wave using only their measured BP and self-reported diagnoses from that wave, we avoided contaminating Wave 2 classifications with Wave 1 measurement errors.

**Table 1. Definitions of hypertension states.**

| | |
|---|---|
| **All hypertension (HTN)** | Systolic BP $\geq$140 mm Hg OR diastolic BP $\geq$ 90 mm Hg |
| | *OR* self-reported to have ever been diagnosed with high BP |
| **HTN Diagnosed** | HTN *AND* self-reported to have ever been diagnosed with high BP |
| **HTN Undiagnosed** | HTN *AND* self-reported never having been diagnosed with high BP |
| **HTN Treated** | HTN *AND* self-reported taking medication or other treatment for high BP in previous 2 weeks |
| **HTN Untreated** | HTN *AND* self-reported not taking medication or other treatment for hypertension in previous 2 weeks |
| **HTN Controlled** | HTN *AND* systolic BP < 140 mm Hg AND diastolic BP <90 mm Hg |
| **HTN Uncontrolled** | HTN *AND* systolic BP $\geq$140 mm Hg OR diastolic BP $\geq$ 90 mm Hg |

**Wealth index.** To examine socioeconomic inequality in hypertension and its diagnosis, treatment, and control, we used a wealth index to proxy socioeconomic status. The index was the first principal component from analysis of each participant's reported possession of household durable assets and financial resources, as well as the building materials, sanitation, and water supply of their house [28]. S1 Table shows the list of variables included in this analysis.

**Covariates.** We examined associations between persistent undiagnosed, untreated, and uncontrolled hypertension and both sociodemographic and lifestyle characteristics that may plausibly have been related to the risk of hypertension or with access to health services that deliver hypertension care. Specifically, we examined associations with sex and age that are risk factors for hypertension and with cohabiting status, rural/urban location, wealth (index), and health insurance that may each be related with access to care [29, 30]. In addition, we examined associations with smoking, alcohol consumption, and Body Mass Index (BMI) that may each be related to hypertension risks [29, 31, 32]. BMI was calculated from height and weight measured by a healthcare professional at the time of the interview. We categorized respondents as: normal weight (BMI <25.0), overweight (25.0–29.9) and obese (>29.9) [33]. Very few respondents had a BMI lower than 20.0 (n = 19), we included them in the "normal weight" category.

## Statistical analyses

We estimated percentages of the cohort aged 50 years and older in 2014 classified as hypertensive in each wave and in both waves. We also estimated percentages of the cohort with undiagnosed, untreated, and uncontrolled hypertension, unconditionally on being classified as hypertensive. We used transition matrices and visual representations to summarize movements between hypertension states from Wave 1 to Wave 2. We also examined how the probability of having uncontrolled hypertension in Wave 2 differed between those who were diagnosed and undiagnosed in Wave 1. We did the same for those treated and untreated in Wave 1.

We measured socioeconomic inequality prevalence of each hypertension state using a concentration index equal to the (scaled) covariance between an indicator of that state and wealth index rank [34]. A positive (negative) value indicated that richer (poorer) individuals were more likely to be in that state.

We estimated probit models of persistent (from Wave 1 to Wave 2) undiagnosed, untreated, and uncontrolled hypertension and used them to obtain averaged marginal effects that indicated by how much the probability of remaining in each of these states for at least 5 years varied with covariates. We also conducted a probit regression to estimate how the probability of transitioning from undiagnosed hypertension in Wave 1 to diagnosed in Wave 2 varied with covariates. The sample used for this analysis consisted of those undiagnosed in Wave

1. We conducted analogous analyses to examine variation in the transition probabilities between untreated and treated and between uncontrolled and controlled.

We did not apply sampling weights since these were not available at cohort level. We assessed representativeness of the cohort by comparing its sociodemographic composition with that of the full Wave 2 cross-sectional sample with sampling weights representative of the population aged 50 years and older in 2014. We took account of stratification and cluster sampling in all statistical interference. STATA 16.0 was used for all analyses.

## Results

### Sample description

Of the 2,998 Wave 2 respondents aged 50 years and older, 1,740 (58%) participated in Wave 1 (Fig 1). In this cohort, valid BP measures were obtained in both waves for 1,254 (72%), and 945 (54%) had full item response on all measures and variables used in the analyses. We present results obtained from the latter, *analysis sample*. Estimates of prevalence and transition rates obtained from the larger sample with BP measures in both waves (*BP sample*) were highly consistent and are given in the S2–S4 Tables.

Table 2 shows characteristics of the analysis sample of Wave 2 respondents aged 50 years and older who participated also in Wave 1 and had full item response. For comparison, the table also shows characteristics of all Wave 2 respondents aged 50 years and older that were weighted to be representative of the Mexican population in that age range [26]. On average, the analysis sample was about eight years older than the full cross-section sample, since new respondents added in Wave 2 were younger than those who had participated in Wave 1. Sample differences in BP and hypertension reflect the difference in average age. Analysis sample respondents were more likely to be rural, have health insurance and abstain from alcohol.

### Hypertension prevalence, diagnosis, treatment and control

Table 3 shows estimates of the prevalence of all hypertension and percentages of the cohort with undiagnosed, untreated, and uncontrolled hypertension in each wave. It also shows estimates of the percent of respondents with these outcomes in both waves. We estimated that 62.4% (95% CI, 58.9 to 65.9) of the cohort was classified as having hypertension in Wave 1. Around five years later when the same respondents were observed in Wave 2, 64.4% (95% CI, 61.0 to 67.7) were classified as having hypertension. The difference between the prevalence rates was not significant (P = 0.364). More than half of the cohort (51.1%; 95% CI, 47.5 to 54.7) was classified as having hypertension in both waves. This percentage is lower than the prevalence in either wave because some respondents (n = 107) transitioned from being classified as hypertensive in Wave 1 to normotensive in Wave 2 (see Table 4). These transitions arise for two reasons. First, measured BP on a single occasion, in a non-clinical setting, can be above the hypertension thresholds in Wave 1 and below the thresholds in Wave 2. If such respondents report in Wave 2 that they have never been diagnosed with high BP/hypertension, then they will not be classified as having hypertension in Wave 2. These cases may have been false positives in Wave 1. Second, a participant could report having ever been diagnosed with hypertension in Wave 1 but in Wave 2 report never having had such a diagnosis. Such reporting implies a measurement error, either in Wave 1 or Wave 2.

We estimated that in Wave 1, 30.3% (95% CI, 27.2 to 33.5) of the cohort had undiagnosed hypertension, 36.0% (95% CI, 32.8 to 39.3) had untreated hypertension, and 55.7% (95% CI, 52.1 to 59.2) had uncontrolled hypertension. In Wave 2, the prevalence rates of undiagnosed, untreated, and uncontrolled hypertension were estimated to be 22.2% (95% CI, 19.7 to 24.9), 27.1% (95% CI, 24.4 to 30.0), and 48.7% (95% CI, 45.4 to 52.0) respectively. Between the two

**Table 2. Sample characteristics, aged 50 years and older years in 2014 (Wave 2).**

| | Analysis sample—observed in Waves 1 & 2 (N = 945) | Comparison sample—observed in Wave 2 (N = 2,998) |
|---|---|---|
| | Mean (SD) | Mean (SD) |
| Age | 70.7 (8.0) | 62.5 (9.3) |
| Systolic blood pressure | 141.8 (23.2) | 138.8 (22.0) |
| Diastolic blood pressure | 76.6 (11.0) | 78.9 (11.0) |
| | No. (%) | No. (%) |
| *Classified as hypertensive* | | |
| Yes | 609 (64.4) | 1,675 (55.9) |
| No | 336 (35.6) | 1,323 (44.1) |
| *Sex* | | |
| Female | 523 (55.3) | 1,613 (53.8) |
| Male | 422 (44.7) | 1,385 (46.2) |
| *Cohabiting* | | |
| Yes | 625 (66.1) | 2,105 (70.2) |
| No | 320 (33.9) | 893 (29.8) |
| *Location* | | |
| Urban | 645 (68.3) | 2,356 (78.6) |
| Rural | 300 (31.8) | 642 (21.4) |
| *Health insurance* | | |
| Yes | 845 (89.4) | 2,508 (83.7) |
| No | 100 (10.6) | 490 (16.3) |
| *Smoker* | | |
| Yes | 104 (11.0) | 375 (12.5) |
| No | 841 (89.0) | 2,623 (87.5) |
| *Drinks alcohol* | | |
| Yes | 438 (46.3) | 1,844 (61.5) |
| No | 507 (53.7) | 1,154 (38.5) |
| *BMI* | | |
| Normal | 270 (28.6) | 670 (23.2) |
| Overweight | 397 (42.0) | 1,236 (41.2) |
| Obese | 278 (29.4) | 1,065 (35.5) |

*Notes*. The analysis sample was observed in both waves and had full item response. The comparison sample was observed in Wave 2. Survey sampling weights were applied to the comparison sample to make it representative of the population aged 50 years and older in 2014.

waves, there was a significant reduction in the prevalence of hypertension that was undiagnosed (P = 0.000), untreated (P = 0.000), and uncontrolled (P = 0.000). Over one-tenth (11.3%; 95% CI, 9.5 to 13.4) were classified as having undiagnosed hypertension in both waves. We estimated that 15.3% (95% CI, 13.2 to 17.8) had untreated hypertension over the five years spanning the two waves, and more than one third (34.7%; 95% CI, 31.7 to 37.9) persistently had uncontrolled hypertension.

## Transitions between hypertension states

Fig 2 and Table 4 show transitions between hypertension states. Panel A shows transitions between three states defined by hypertension and diagnosis. Of the 355 respondents (229+48+78)

**Table 3. Prevalence of hypertension and undiagnosed, untreated, and uncontrolled hypertension.**

| | N = 945 | | | | | |
| --- | --- | --- | --- | --- | --- | --- |
| | Wave 1 | | Wave 2 | | Both waves | |
| | No. | (%) [95% CI] | No. | (%) [95% CI] | No. | (%) [95% CI] |
| All hypertension (HTN) | 590 | (62.4) [58.9–65.9] | 609 | (64.4) [61.0–67.7] | 483 | (51.1) [47.5–54.7] |
| HTN Undiagnosed | 286 | (30.3) [(27.2–33.5] | 210 | (22.2) [19.7–24.9] | 107 | (11.3) [9.5–13.4] |
| HTN Untreated | 340 | (36.0) [32.8–39.3] | 256 | (27.1) [24.4–30.0] | 145 | (15.3) [13.2–17.8] |
| HTN Uncontrolled | 526 | (55.7) [52.1–59.2] | 460 | (48.7) [45.4–52.0] | 328 | (34.7) [31.7–37.9] |

who did not have hypertension in Wave 1, 13.5% (95% CI, 10.4 to 17.4) were classified with hypertension and had been diagnosed by Wave 2. A larger percentage (22%; 95% CI, 18.3 to 26.1) of those initially not classified with hypertension were classified as having the condition in Wave 2 but had not been diagnosed. This means that more than three fifths (62% = 22/(13.5+22)) of those who became hypertensive were undiagnosed. Of the 286 respondents

**Table 4. Transitions between hypertension states.**

| A. HTN Diagnosed | Wave 2 | | |
| --- | --- | --- | --- |
| Wave 1 | No HTN | HTN Diagnosed | HTN Undiagnosed |
| | No. (%) | No. (%) | No. (%) |
| | [95% CI] | [95% CI] | [95% CI] |
| No HTN | 229 (64.5) | 48 (13.5) | 78 (22.0) |
| | [59.9, 68.8] | [10.4, 17.4] | [18.3, 26.1] |
| HTN Diagnosed | 22 (7.2) | 257 (84.5) | 25 (8.3) |
| | [4.9, 0.6] | [80.2, 88.1] | [5.6, 11.9] |
| HTN Undiagnosed | 85 (29.7) | 94 (32.9) | 107 (37.4) |
| | [24.3, 35.7] | [27.8, 38.3] | [32.7, 42.4] |
| **B. HTN Treated** | **Wave 2** | | |
| Wave 1 | No HTN | HTN Treated | HTN Untreated |
| | No. (%) | No. (%) | No. (%) |
| | [95% CI] | [95% CI] | [95% CI] |
| No HTN | 229 (64.5) | 41 (11.6) | 85 (23.9) |
| | [59,9–69,8] | [8.6, 15.4] | [20.2, 28.2] |
| HTN Treated | 18 (7.2) | 206 (82.4) | 26 (10.4) |
| | [4.9, 10.6] | [77.3, 86.6] | [7.3, 14.6] |
| HTN Untreated | 89 (26.2) | 106 (31.2) | 145 (42.7) |
| | [21.5, 31.5] | [26.6, 36.2] | [37.9, 47.5] |
| **C. HTN Control** | **Wave 2** | | |
| Wave 1 | No HTN | HTN Controlled | HTN Uncontrolled |
| | No. (%) | No. (%) | No. (%) |
| | [95% CI] | [95% CI] | [95% CI] |
| No HTN | 229 (64.5) | 24 (6.8) | 102 (28.7) |
| | [59.9, 68.8] | [4.7, 9.7] | [24.7, 33.2] |
| HTN Controlled | 6 (9.4) | 28 (43.8) | 30 (46.9) |
| | [4.4, 19.0] | [31.3, 57.0] | [33.7, 60.5] |
| HTN Uncontrolled | 101 (19.2) | 97 (18.4) | 328 (62.4) |
| | [15.6, 23.4] | [15.3, 22.0] | [57.8, 66.7] |

Notes. The first number in each cell is a frequency. The second is the row percent in that cell. In parentheses is the 95% confidence interval for the row percent.

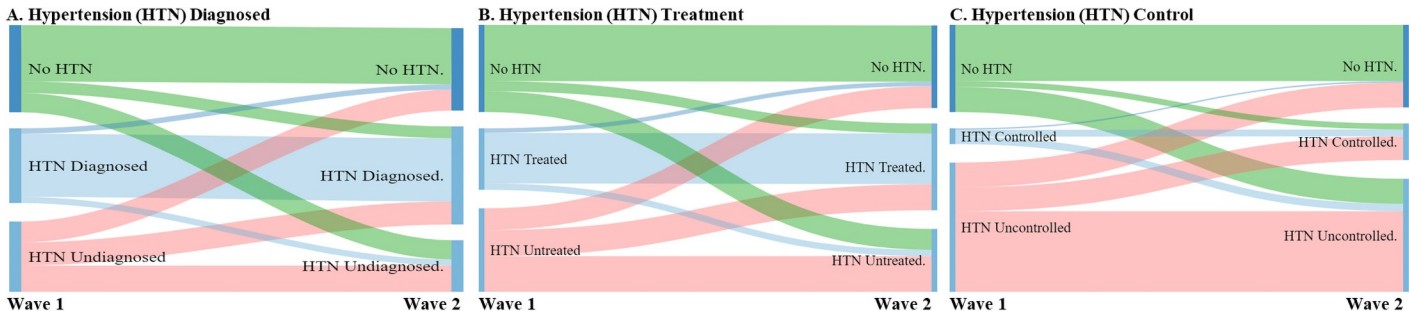

**Fig 2. Transitions between hypertension states.** *Notes*. Each figure in the top panel gives a visual representation of the data presented in the respective transition matrix table below it. The sample size is 945 for each panel.

(85+94+107) who were classified as having hypertension in Wave 1 but reported never having been diagnosed, 37.4% (95% CI, 32.7 to 42.4) remained undiagnosed five years later, while 32.9% (95% CI, 27.8 to 38.3) acquired a diagnosis and 29.7% (95% CI, 24.3 to 35.7) were reclassified, on the basis of measured BP and reported diagnosis, as not being hypertensive in Wave 2. A small fraction (8.3%; 95% CI, 5.6 to 11.9) of the 304 respondents (22+257+25) who were classified as having hypertension and reported ever having been diagnosed in Wave 1 had BP above the hypertension thresholds in Wave 2 but at that time they reported, inconsistently, that they had never been diagnosed.

Panel B shows that 11.6% (95% CI, 8.6 to 15.4) of those not classified with hypertension in Wave 1 were classified as having hypertension and in receipt of treatment in Wave 2, while 23.9% (95% CI, 20.2 to 28.2) were reclassified as having untreated hypertension. There was considerable persistence in treatment: 82.4% (95% CI, 77.3 to 86.6) of those who were being treated for hypertension in Wave 1 continued to be in treatment five years later. Over one tenth (10.4%; 95% CI, 7.3 to 14.6) of those initially under treatment were no longer treated in Wave 2 but were still classified as having hypertension. A small but sizeable percentage (7.2%; 95% CI, 4.9 to 10.6) of those who were being treated in Wave 1 were classified as not having hypertension in Wave 2, which implies that they reported in that wave, inconsistently with their previous reported treatment, never having been diagnosed with hypertension. More than two fifths (42.7%; 95% CI, 37.9 to 47.5) of those initially classified as having untreated hypertension were still untreated. Almost a third (31.2%; 95% CI, 26.6 to 36.2) of those with untreated hypertension in Wave 1 were under treatment in Wave 2.

Panel C reveals that 28.7% (95% CI, 24.7 to 33.2) of those who were free of a hypertension classification in Wave 1 had uncontrolled hypertension five years later. Among the relatively small number (64 = 6+28+30) identified as having controlled hypertension in Wave 1, almost half (46.9%; 95% CI, 33.7 to 60.5) moved to uncontrolled hypertension. Among the much larger number (526 = 101+97+328) who had uncontrolled hypertension in Wave 1, 62.4% (95% CI, 57.8 to 66.7) were still in this state in Wave 2, while only 18.4% (95% CI, 15.3 to 22.0) achieving BP control. Further analyses revealed that those with diagnosed hypertension in Wave 1 were more than twice as likely as those initially undiagnosed to have their BP controlled in Wave 2 (S5 Table). There was a similar difference between those initially treated and untreated in their relative likelihoods of achieving BP control by the end of the study period.

## Socioeconomic inequality

Table 5 shows concentration indices that measure wealth-related inequality in each hypertension indicator. The positive concentration indices imply that a hypertension classification was

**Table 5. Concentration indices of all hypertension and undiagnosed, untreated, and uncontrolled hypertension.**

| | N = 945 | | |
| --- | --- | --- | --- |
| | Wave 1 | Wave 2 | Both waves |
| | Concentration index [95% CI] | | |
| All hypertension (HTN) | 0.007 [-0.06, 0.08] | 0.026 [-0.04, 0.09] | 0.028 [-0.04, 0.10] |
| HTN Undiagnosed | -0.039 [-0.11, 0.02] | -0.026 [-0.09, 0.03] | -0.007 [-0.05, 0.04] |
| HTN Untreated | -0.068 [-0.14, 0.00] | -0.039 [-0.11, 0.03] | -0.050 [-0.10, 0.00] |
| HTN Uncontrolled | -0.029 [-0.10, 0.04] | -0.029 [-0.10, 0.04] | -0.047 [-0.11, 0.02] |

more prevalent among wealthier respondents in each wave and that wealthier respondents were more likely to be classified as hypertensive in both waves. However, all of the 95% confidence intervals include zero so there was no evidence of statistically significant inequality in hypertension prevalence. The next three rows of the table show concentration indices for undiagnosed, untreated, and uncontrolled hypertension in each wave and in both waves. All the point estimates of these concentration indices are negative, indicating that poorer respondents were more likely to have undiagnosed, untreated, and uncontrolled hypertension. However, all of the 95% confidence intervals include zero, and thus the inequality apparent in the sample was not statistically significant.

## Multivariable analysis

Table 6 contains results of respondents' characteristics regressed on having persistent undiagnosed, untreated and uncontrolled hypertension in both waves. Men had a 10 percentage point (pp) higher probability of remaining undiagnosed. They were also significantly more likely than women to remain untreated (by 9 pp, with p-value <0.05) and uncontrolled (by 3 pp), although the 95% CI for the latter estimate includes zero. Singles were significantly more likely to remain undiagnosed (4 pp, with p-value <0.05) compared to cohabiting respondents. Rural dwellers in the sample were more likely to have persistent undiagnosed, untreated, and uncontrolled hypertension, although only their estimated 9 pp higher probability of remaining uncontrolled has a 95% CI that does not include zero. Those without health insurance were 4 pp more likely to remain undiagnosed (significant with p-value < 0.05). Those with overweight were significantly more likely to remain untreated or uncontrolled in both waves, respectively with 5 and 13 pp. Compared with abstainers, consumers of alcohol were less likely to remain undiagnosed (3 pp) and untreated (4 pp) with a p-value < 0.05. Analyses of variation in the probabilities of transitioning from undiagnosed to diagnosed, untreated to treated, uncontrolled to controlled revealed that men were significantly less likely to make each of these transitions (S6 Table). Smokers were significantly more likely to move from undiagnosed to diagnosed.

## Discussion

This study is among the first to provide a longitudinal perspective on diagnosis, treatment, and control of hypertension [35, 36] in a middle-income country. We used data from a cohort of Mexicans aged 50 years and older in two waves of the WHO Study on Global AGEing and adult health (SAGE) collected in 2009 and 2014. We found a substantial prevalence of hypertension (64%). Prevalence of undiagnosed, untreated, and uncontrolled hypertension significantly decreased over the five year period to reach 22%, 27%, and 49%, respectively. More than one third of those classified as having undiagnosed hypertension were still in this state five years later, more than two fifths of those initially untreated remained untreated, and over three

**Table 6. Averaged marginal effects on probabilities of persistent undiagnosed, untreated, and uncontrolled hypertension.**

| | All respondents (N = 945) | | |
| --- | --- | --- | --- |
| | **HTN Undiagnosed in both waves** | **HTN Untreated in both waves** | **HTN Uncontrolled in both waves** |
| | ME [95% CI] (P-value) | ME [95% CI] (P-value) | ME [95% CI] (P-value) |
| **Sex** | | | |
| Female | Ref | Ref | Ref |
| Male | 0.10 [0.06, 0.15] (0.000) | 0.09 [0.03, 0.14)] (0.002) | 0.03 [-0.04, 0.11] (0.366) |
| **Age (years)** | -0.00 [-0.00, 0.00] (0.900) | 0.00 [-0.00, 0.00] (0.068) | 0.01 [0.01, 0.01] (0.001) |
| **Cohabiting** | | | |
| Yes | Ref | Ref | Ref |
| No | 0.04 [0.01, 0.07] (0.007) | 0.05 [-0.00, 0.07] (0.061) | 0.06 [-0.01, 0.12] (0.107) |
| **Living area** | | | |
| Urban | Ref | Ref | Ref |
| Rural | 0.02 [-0.00, 0.05] (0.083) | 0.03 [-0.1, 0.07] (0.178) | 0.09 [0.02, 0.16] (0.013) |
| **Health insurance** | | | |
| Health insurance | Ref | Ref | Ref |
| No health insurance | 0.04 [0.01, 0.08] (0.046) | 0.05 [-0.00, 0.10] (0.103) | 0.07 [-0.03, 0.16] (0.171) |
| **Wealth status** | | | |
| Tercile 1 | Ref | Ref | Ref |
| Tercile 2 | -0.01 [-0.04, 0.03] (0.710) | -0.02 [-0.06, 0.03] (0.431) | -0.03 [-0.11, 0.04] (0.411) |
| Tercile 3 | 0.02 [-0.02, 0.05] (0.334) | -0.02 [-0.01, 0.03] (0.480) | -0.01 [-0.09, 0.07] (0.766) |
| **Body weight** | | | |
| Normal weight | Ref | Ref | Ref |
| Overweight | 0.02 [-0.01, 0.05] (0.284) | 0.05 [0.01, 0.10] (0.023) | 0.13 [0.05, 0.20] (0.001) |
| Obese | -0.00 [-0.04, 0.04] (0.872) | 0.04 [-0.01, 0.08] (0.134) | 0.16 [0.08, 0.25] (0.000) |
| **Smoker** | | | |
| No | Ref | Ref | Ref |
| Yes | 0.00 [-0.04, 0.04] (0.931) | 0.02 [-0.04, 0.08] (0.480) | 0.07 [-0.03, 0.17] (0.182) |
| **Alcohol consumption** | | | |
| No | Ref | Ref | Ref |
| Yes | -0.03 [-0.06, -0.01] (0.017) | -0.04 [-0.08, -0.00] (0.048) | -0.04 [-0.11–0.03] (0.250) |

*Note.* Probit estimates of marginal effects averaged over the respective samples.

fifths of those initially with uncontrolled hypertension failed to achieve BP control by the end of the period. The likelihood of experiencing continued uncontrolled hypertension was much higher than the chances of achieving BP control, which signals substantial losses in population health since CVD risks rise steeply with the duration of exposure to uncontrolled hypertension [37]. These estimates confirm substantial persistence of unfavorable hypertension states, ongoing failures of the health system to find patients who had fallen through the cracks of hypertension care, and lack of patient adherence to treatment. We cannot claim that these findings would necessarily extend beyond Mexico. They may, however, motivate estimation of the prevalence of persistent undiagnosed, untreated, and uncontrolled hypertension in other countries.

We are aware of two other studies that took a longitudinal approach. One study conducted in Ghana, did not assess transitions between the hypertension states but did report similar factors associated with hypertension diagnosis i.e. residing in urban areas and having health insurance [36]. The other (preprint) study is a multi-country including Mexico with similarities to our longitudinal design [35]. The transition rates from undiagnosed to diagnosed, and

untreated to treated are like in our study close to 30%, while we find a tree times higher rate for treatment continuity. In line with our work, they find men and rural dwellers to be less likely to advance forward through the continuum of hypertension care. The difference in treatment continuity might be driven by differences in the average characteristics of both cohorts. The cohort used by Mauer et al. [35] is derived from the Mexican Family Life Survey which also includes those aged 40 to 49 years old and had it first wave a few years earlier (2005). The second wave was apparently collected over a prolonged period of time (2009 till 2012). Those timing differences might have resulted in a lower observed treatment continuity given that the cohort was followed over a longer and less strictly defined time period.

Our approach allowed for reclassifications from hypertensive to normotensive between waves and we found that such transitions are far from uncommon. Approximately, these are as common as moving to a diagnosed, treated, or controlled state. They do not derive from a false premise that someone with clinically diagnosed hypertension can be cured. In this study, a participant could have been reclassified as not having hypertension because their BP fell from being above the hypertension thresholds when measured (on a single occasion) at Wave 1 to below these thresholds at Wave 2 and they reported never having been diagnosed with hypertension at Wave 2. Reclassification could also occur if the participant never had BP above the thresholds but inconsistently reported having been diagnosed with hypertension at Wave 1 but never having been diagnosed at Wave 2. Each reason for reclassification derives from a measurement error—a false positive in the first case, inconsistent reporting of diagnosis in the second—that would bias cross-sectional estimates of hypertension diagnosis, treatment, and control. While these errors suggest that cross-sectional studies have likely overestimated rates of undiagnosed, untreated, and uncontrolled hypertension, this is not sufficient reason for less policy concern about these indicators of gaps in hypertension screening and management.

We compared how the probability of achieving BP control differed between those who had been diagnosed five years earlier and those who had not. The initially diagnosed were more than twice as likely to have controlled BP after five years. This supports the case for effective implementation of opportunistic or population-based screening for hypertension. The rate of persistent untreated hypertension was high and the initially treated were more than twice as likely as the untreated to have achieved BP control after five years. This points to the need for improvements in hypertension management, as well as screening. The potential health gains from such improvements are clear (37) given evidence that antihypertensives are highly cost-effective (38), as are lifestyle changes if they can be achieved. There was a high degree of persistence in treatment: more than four fifths of those who were under treatment at the beginning of the period continued with treatment five years later. Taken together, these results suggest that diagnosing people and getting them on treatment is the primary challenge, while maintaining continuity of care is arguably of a secondary order. That said, multiple studies have shown that half of patients prescribed antihypertensives stopped taking them within a year [38–40]. Lack of treatment adherence is a recognized global concern [41]. The high rate of persistent uncontrolled hypertension we find provides further support for making frequent follow-up of patients who have not achieved BP control a key component of a healthcare team's concerted effort to improve adherence [42].

In the study cohort, hypertension was slightly more prevalent among the wealthier. This adds to already conflicting evidence from Latin America regarding socioeconomic inequality since it is reported that individuals with a lower SES had a higher risk for an elevated blood pressure, while another study summarizes recent evidence from LMIC settings with the majority of the studies confirming the positive relationship between socioeconomic status and chronic conditions (including hypertension) [43, 44]. Furthermore, evidence from a low-income setting in Mexico revealed that using two different aspects of SES showed an inverse

association with elevated blood pressure [45]. In our sample, we found that less wealthy individuals were slightly more likely to have persistent undiagnosed hypertension and more likely to have persistent untreated and uncontrolled hypertension, however, these differences were not significant. Previous evidence showed that the performance of health systems in LMICs regarding the management of hypertension was poor: not even halve of those with hypertension were diagnosed, only one third were taking medication and 10% had their blood pressure under control [46]. Moreover, individuals with a lower household wealth were more likely to be lost to care before reaching the phase of blood pressure control [46]. The fact that, at least in the sample, the wealthier were more likely to have hypertension but less likely to have undiagnosed (as well as untreated and uncontrolled) hypertension suggests that the former positive wealth gradient in hypertension prevalence is partly due to the wealthier being more likely to get diagnosed. We found that, compared with abstainers, alcohol consumers were less likely to remain undiagnosed. The rate of alcohol consumers in our sample was lower compared to the comparison sample, therefore we have difficulties with interpreting this finding. Furthermore, we found that in the sample, men, those living alone, rural dwellers, uninsured, and those with overweight were more likely to have persistent undiagnosed, untreated, and uncontrolled hypertension. These sociodemographic groups appeared to have been most exposed to deficiencies in hypertension screening and management, and possibly most laxed in adherence to treatment. Other studies, though cross-sectional, observe similar characteristics (e.g. having health insurance, educated, married, living area) for individuals who were less likely to have (undiagnosed, untreated, uncontrolled) hypertension [17, 18, 47]. Previous evidence suggested that enrollees in Mexico's flagship Seguro Popular universal coverage program had better access to health care, including diagnosis and treatment of hypertension [48, 49]. In line with this, we found that sample respondents that did not have health insurance were more likely to experience persistent undiagnosed, untreated, and uncontrolled hypertension, although this was only statistically significant for persistent undiagnosed hypertension. Finally, we found that women are more likely to become diagnosed, treated and controlled, and thus receive diagnoses or treatment or reach controlled hypertension.

## Limitations

We restricted the sample to respondents who responded to both Wave 1 and Wave 2. The low response rate in Wave 1, as well as attrition between waves, potentially made the study cohort unrepresentative of the Mexican population aged 50 years and older at the time of Wave 2 (2014). Comparison with the Wave 2 cross-section sample weighted to be representative of the population aged 50 years and older showed that the cohort was older, and, consequently, had higher rates of hypertension, rural dwellers, and health insurance coverage, and it was less likely to be cohabiting and to drink alcohol. Our results should be interpreted with these differences in mind. They do not necessarily hold for the population of Mexico aged 50 years and older in 2014, although they are likely to be more representative for an older population. Selective attrition could also potentially leave the cohort unrepresentative with respect to unobserved characteristics that are related to hypertension and its management.

Respondents who had an elevated BP reading in Wave 1 were informed of this and advised to seek medical advice. Consequently, we would expect rates of persistent undiagnosed, untreated, and uncontrolled hypertension to have been lower in the study cohort than they were in the population. For this reason, the high rates we found are of even greater concern.

The main limitation of this and most hypertension awareness, treatment, and control studies is that BP was measured on a single occasion in each wave. While it was measured multiple times on one occasion, it would have been better if there was a longer time between these two

periods. Hypertension is usually diagnosed from BP measurements made on at least two occasions. This might have increased the number of false positives among those identified as having hypertension. The true rate of undiagnosed hypertension in each wave—not persistent undiagnosed hypertension between waves—is likely to be lower than estimated. However, the longitudinal perspective taken in this study provided insight into this measurement error problem that is missing from cross-sectional studies. We estimated that 30% (n = 85) of those identified as having undiagnosed hypertension in Wave 1 were identified as not having hypertension in Wave 2. The respective rates for untreated and uncontrolled hypertension were 26% (n = 89) and 19% (n = 101). These estimates suggest that false positive may well cause substantial upward bias in cross-section estimates of these rates [50]. The focus of this study was not on a cross-sectional snapshot but on persistent gaps in hypertension diagnosis and management over a 5-year period. Classification errors, while still present, are less problematic from this longitudinal perspective.

Our study covered the period 2009–2014. Since then, the Mexican Institute of Social Security (IMSS) has tried to tilt its model of care towards prevention [51] and has introduced several integrated programs of care [52, 53]. It could be that these policies have improved hypertension screening and management and corrected some of the care deficiencies suggested by our estimates.

## Policy implications

Our estimates of substantial rates of persistent undiagnosed, untreated, and uncontrolled hypertension suggest that clinical and public health interventions are required to improve hypertension screening and care. A regular BP check during healthcare visits for other conditions may lead to more and earlier diagnoses. Our results show that this could be particularly relevant for those who are male, single, rural dwellers, uninsured or overweight. The substantial rate of transition from controlled to uncontrolled hypertension suggests that policies to improve treatment adherence care continuity would be particularly valuable. Association of persistent undiagnosed hypertension with lack of health insurance suggests that improving effective coverage for primary care, or even just making people aware of their insurance entitlement, may help close gaps in hypertension care.

## Conclusions

Our study showed that a large proportion of the Mexican older population with hypertension remained undiagnosed, untreated, and uncontrolled for at least five years and that these hypertensive stages have a dynamic character. We show that there is room for improvement in hypertension diagnosis, long-term treatment adherence and hypertension control. To ensure more equitable hypertension management and effectively prevent premature deaths, increased diagnosis and long-term treatment efforts should be directed towards men, those living alone, rural dwellers, uninsured and those with overweight.

## Supporting information

**S1 Table. Variables included in this principal component analysis for the wealth index.**
(DOCX)

**S2 Table. Characteristics of Wave 2 respondents aged 50 years and older years in 2014.**
(DOCX)

**S3 Table. Prevalence of hypertension and unaware, untreated, and uncontrolled hypertension, incomplete case sample.**
(DOCX)

**S4 Table. Transitions between hypertension states between Wave 1 and Wave 2, incomplete case sample.**
(DOCX)

**S5 Table. Transitions from hypertension diagnoses and treatment to control, respondents with hypertension in both waves.**
(DOCX)

**S6 Table. Averaged marginal effects (ME) on probabilities of transitioning to diagnosed, treated, and controlled hypertension.**
(DOCX)

## Author Contributions

**Conceptualization:** C. M. Dieteren, O. O'Donnell, I. Bonfrer.

**Data curation:** C. M. Dieteren.

**Formal analysis:** C. M. Dieteren.

**Investigation:** C. M. Dieteren.

**Methodology:** C. M. Dieteren, O. O'Donnell, I. Bonfrer.

**Project administration:** C. M. Dieteren.

**Supervision:** O. O'Donnell, I. Bonfrer.

**Validation:** O. O'Donnell, I. Bonfrer.

**Visualization:** C. M. Dieteren.

**Writing – original draft:** C. M. Dieteren.

**Writing – review & editing:** O. O'Donnell, I. Bonfrer.

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
