## [Decision Letter · Decision Letter 0]

25 Jun 2021

 PGPH-D-21-00021 Prevalence and inequality in persistent undiagnosed, untreated, and uncontrolled hypertension: longitudinal analysis of a cohort of older Mexicans PLOS Global Public Health

Dear Dr. Dieteren,

Thank you for submitting your manuscript to PLOS Global Public Health. After careful consideration, we feel that it has merit but does not fully meet PLOS Global Public Health’s publication criteria as it currently stands. Therefore, we invite you to submit a revised version of the manuscript that addresses the points raised during the review process.

We look forward to receiving your revised manuscript.

Kind regards,

Biplab Kumar Datta, Ph.D.

Academic Editor

Journal Requirements:

Additional Editor Comments (if provided):

There is a conceptual problem in determining the hypertensive individuals in the study. Hypertension is a chronic condition that may not be cured but can be controlled. Therefore, an individual, once (e.g., wave-1) diagnosed as hypertensive can either have controlled or uncontrolled hypertension; but can not be categorized as not having hypertension at a later period (e.g., wave-2). Hence, the outcome variable, “HTN in both waves”, is technically incorrect. All hypertensive individuals in wave-1, should be categorized as hypertensive in wave-2 as well. Some of them may have hypertension under control in wave-2 either by taking anti-hypertensive medication or adopting healthy lifestyle (e.g., physical exercise, food habit, etc.) or both. Please address this issue and revise the analysis as required.

I personally think the advantage of having a longitudinal data was under-utilized in achieving the aim of the study (i.e., identifying the socioeconomic characteristics associates with hypertension care cascade - diagnosed, treated, controlled). For example, the analyses do not offer what share of undiagnosed hypertensive individuals in wave-1 were diagnosed in wave-2 and how that differ by the respondent's sociodemographic characteristics. Similarly, those were diagnosed but not treated in wave-1, how their sociodemographic characteristics were associated with receiving treatment in wave-2. As pointed out by Reviewer 1, please emphasis on what the longitudinal analyses revealed on this issue. It appears that the longitudinal data were treated as repeated cross-sectional data. Please clarify and justify.

Please clarify whether the complex survey weights were used in the analysis, if not then provide justification.

Provide a discussion on how the findings of this paper are comparable with hypertension care in other countries.

Reviewers' comments:

Reviewer's Responses to Questions

**Comments to the Author**

1. Does this manuscript meet PLOS Climate’s publication criteria? Is the manuscript technically sound, and do the data support the conclusions? The manuscript must describe methodologically and ethically rigorous research with conclusions that are appropriately drawn based on the data presented.

Reviewer #1: Yes

Reviewer #2: Partly

2. Has the statistical analysis been performed appropriately and rigorously?

Reviewer #1: No

Reviewer #2: No

3. Have the authors made all data underlying the findings in their manuscript fully available (please refer to the Data Availability Statement at the start of the manuscript PDF file)?

Reviewer #1: Yes

Reviewer #2: Yes

4. Is the manuscript presented in an intelligible fashion and written in standard English?

PLOS Climate does not copyedit accepted manuscripts, so the language in submitted articles must be clear, correct, and unambiguous. Any typographical or grammatical errors should be corrected at revision, so please note any specific errors here.

Reviewer #1: No

Reviewer #2: Yes

5. Review Comments to the Author

Reviewer #1: I have attached my comments as a separate document.

Reviewer #2: The study aims to estimate prevalence and socioeconomic inequalities in persistent undiagnosed, untreated, and uncontrolled hypertension using waves 1 (2009/10) and 2 (2014) of the WHO Study on Global AGEing and adult health (SAGE) that followed a cohort of Mexicans aged 50+ years.

Strengths:

Longitudinal dataset

My comments are below

Introduction:

Not sure what is the context of the reference of undiagnosed hypertension discussed in the first paragraph in the introduction. Is it in the context of developing countries or developed countries?

Introduction needs further motivation regarding the role of socioeconomic inequalities in undiagnosed, untreated, and uncontrolled hypertension in the population of interest. It seems that role of socioeconomic inequity is key here but no attention has been paid in the introduction to motivate why SES inequity could contribute to undiagnosed, untreated, and uncontrolled hypertension in this population.

Method:

What is the final sample size that includes both wave 1 and Wave 2 respondents?

Wave 1 response rate seems to be really low (51%); did author perform any test that nonresponse rate in Wave 1 could contribute to bias in the final sample? In other words, are there any differences in individual characteristics between those who responded in wave 1 and those who did not?

Results:

From Table 2, it seems that there are some differences in prevalence of hypertension and other individual characteristics (location, drinking alcohol etc. ) between analytic sample that combined both wave and 2 and comparison sample wave 2 only. Why is that? Authors need to offer implications of these differences and how that could contribute to the results?

Is there any statistically significant difference between prevalence of hypertension in wave 1 and wave 2?

Table 5 needs some additional clarification—not sure if I follow the relationship between socioeconomic inequality and hypertension prevalence as author discussed on page 12

What is the outcome variable of the multivariate analysis (hypertension diagnosed in both waves)?

Authors need to offer clear definition of the outcome measure before discussing multivariate analysis. Also, what was rationale of having this outcome that having hypertension in both waves? In general, if some has hypertension , it will likely be the case that after 5 years that person will most likely to have hypertension as well. It will be rather interesting to see, if someone is undiagnosed or not diagnosed with hypertension in wave 1, what are the likelihood that the person will be diagnosed with hypertension in wave 2 or remain undiagnosed (although had it) in wave 2?

What is “pp?” I assume it means “percentage point”?

Again for the other outcome variables also some combination between wave would be interesting to examine. Also, please provide p-values in addition to CI for the probit regression results and discuss statistical significance of the results.

It was not clear why only a handful of control variables were measured in the analysis, I believe the survey includes a host of control variables. Need description of these variables as well. For example, marital status, education etc. why not controlled in the analysis?

Discussion: currently discuss section seem to be the repetition of the results section, I would suggest author to discuss in the context of current literature and implications for clinical decision-making and health outcomes.

6. PLOS authors have the option to publish the peer review history of their article (what does this mean?). If published, this will include your full peer review and any attached files.

**Do you want your identity to be public for this peer review?** For information about this choice, including consent withdrawal, please see our Privacy Policy.

Reviewer #1: No

Reviewer #2: No

---

## [Editor Report · Decision Letter 1]

31 Aug 2021

 PGPH-D-21-00021R1 Prevalence and inequality in persistent undiagnosed, untreated, and uncontrolled hypertension: longitudinal analysis of a cohort of older Mexicans PLOS Global Public Health

Dear Dr. Dieteren,

Thank you for submitting your manuscript to PLOS Global Public Health. After careful consideration, we feel that it has merit but does not fully meet PLOS Global Public Health’s publication criteria as it currently stands. Therefore, we invite you to submit a revised version of the manuscript that addresses the points raised during the review process.

We look forward to receiving your revised manuscript.

Kind regards,

Biplab Kumar Datta, Ph.D.

Academic Editor

Journal Requirements:

Additional Editor Comments (if provided):

Thank you for addressing the concerns raised by the reviewers. However, it appears that the longitudinal analysis, which is core to this paper, requires further improvement. The introduction was heavily built on the necessity of the longitudinal analysis to better understand the status of the hypertension management and assessing the needs accordingly. The analyses, however, did not adequately exploit the longitudinal techniques. For example, figure 2 and tables 3 and 4 basically provide descriptive statistics observed over time. A proper longitudinal analysis would require estimating panel models to report how odds of being diagnosed, treated, or having hypertension under control changes over time. A simple way of doing that is to estimate a panel probit model with a year dummy (e.g., d=1 if wave 2 and 0 if wave 1). The coefficient (or avg. marginal effect) of the year dummy would capture the change across waves. An advanced model could be panel multinomial logit with mutually exclusive outcomes such as undiagnosed, diagnosed but not treated, treated but not controlled, and controlled. Unfortunately as I mentioned previously, without any proper longitudinal empirical analysis, the longitudinal perspective story is not viable. Similar is the case for the longitudinal analysis with respect to sociodemographic characteristics. The panel structure of the data is merely utilized when a probit model is estimated rather than a panel model that would exploit the within person change over time. Table 6 is a cross-section and repeated cross-section analysis and does not provide any longitudinal perspective. Table S6 is not a proper longitudinal analysis as it does not apply required empirical techniques to offer insights on within or between changes. Please revise the analysis utilizing proper longitudinal techniques.
---

## [Editor Report · Decision Letter 2]

21 Oct 2021

PGPH-D-21-00021R2

Prevalence and inequality in persistent undiagnosed, untreated, and uncontrolled hypertension: evidence from a cohort of older Mexicans

Dear Dr. Dieteren,

Thank you for submitting your manuscript to PLOS Global Public Health. After careful consideration, we feel that it has merit but does not fully meet PLOS Global Public Health’s publication criteria as it currently stands. Therefore, we invite you to submit a revised version of the manuscript that addresses the points raised during the review process.

We look forward to receiving your revised manuscript.

Kind regards,

Biplab Kumar Datta, Ph.D.

Academic Editor

Journal Requirements:

Additional Editor Comments (if provided):

I thank the authors to clarify their position on the analyses and scopes of the paper. There were indeed some different expectations based on the part of introduction that states limitations of using cross-sectional data and benefits of using longitudinal data in the introduction section as follows:

The introduction states, “First, it does not inform of how long hypertension remains undiagnosed, untreated, and uncontrolled.” – answering this question warrants survival type analysis, which was not an analysis offered in this paper, nor would that been feasible given the nature of data. What this paper offers are how many of the undiagnosed patients remained undiagnosed after five years, or how many of the untreated remained untreated after five years, which is very different from how long hypertension remained undiagnosed or untreated.

The introduction states, “Second, with cross-sectional data one cannot determine the rates at which people move from undiagnosed to diagnosed, untreated to treated, and uncontrolled to controlled, and how these transition probabilities vary with sociodemographic characteristics.” – the standard expectation in this regard, particularly from the policy perspective, is to know how the odds of transition were associated with certain sociodemographic characteristics. While the paper provides simple transition rates, it does not offer how the actual transition rates differs across sociodemographic groups. The multivariate analysis presented in table 6 does not tell us how x% transition rate from wave-1 to wave-2 varies across wealth status or across urban and rural areas.

The introduction states, “Third, analyses of cross-sectional data are particularly prone to finding false positive cases of hypertension as a result of relying on blood pressure measurement on a single occasion. These false positives will bias cross-section estimates of undiagnosed, untreated, and uncontrolled hypertension upwards.” – this paper clearly does not solve the problem of false positive outcomes as hypertension were not clinically diagnosed and there are overlapping across the two waves in determination of hypertensive status.

Then it states, “All three of these limitations can be addressed with longitudinal data.” – building the expectation that the analyses presented in this paper will address all these shortcomings of a typical cross-sectional analysis of hypertension care cascade. The analyses, however, did not adequately encompass any of the advantages mentioned about longitudinal data in the introduction.

After a long presentation on the benefits of using longitudinal data, the aim of the study was stated as “to estimate the prevalence and socioeconomic distribution of hypertension that remained undiagnosed, untreated, and uncontrolled for at least five years among Mexicans aged 50 years and older and to estimate rates of transition from those states to diagnosis, treatment, and control.” – which didn’t offer significant improvement to any of the “three limiting aspects”.

I appreciate your explanations and do agree with your views regarding the selective scope of the study aims and their execution. However, I feel certain segments of the paper are not aligned with this aim and requires revision. In the context of the stated aim of this paper, I will be happy recommending this article for publication given the following concerns are addressed:

1. As the authors clarify that they are only interested in “estimating the prevalence of persistent undiagnosed, untreated, and uncontrolled hypertension”, the long presentation on the benefits of using longitudinal data in the introduction is unnecessary and misleading to the readers. I, therefore, suggest dropping the two paragraphs (second and third from last) in the introduction as that misleads the readers with very different expectations and is not consistent with the stated aim of the paper.

2. Please make it clear in the introduction that you are interested in “prevalence of persistent undiagnosed/untreated/uncontrolled hypertension”, not in “change in the prevalence of undiagnosed/untreated/uncontrolled hypertension”.

3. I suggest dropping or rephrasing the following sentence in the discussion section: “They do, however, provide sufficient cause to conduct similar analyses of longitudinal data in other countries to determine the extent to which undiagnosed, untreated, and uncontrolled hypertension are transient or persistent states and using this to identify policy priorities.” – since none of the limitations of cross-sectional analysis are substantially improved in this paper (as per the choice of parameters or interest), “similar analyses” of longitudinal data in other countries may not add much to the literature.

4. I suggest dropping or rephrasing the following sentence in the policy implications sub-section: “The longitudinal character of this study allows additional policy recommendations besides the implications one would derive from cross-sectional analyses.” Given the aim of this paper, very similar results (e.g., table 3) could have been produced using repeated cross-section data. I would suggest discussing the policy implications from the persistence (of undiagnosed/ untreated/ uncontrolled) perspective only.
---

## [Editor Report · Decision Letter 3]

18 Nov 2021

Prevalence and inequality in persistent undiagnosed, untreated, and uncontrolled hypertension: evidence from a cohort of older Mexicans

PGPH-D-21-00021R3

Dear Dr. Dieteren,

We're pleased to inform you that your manuscript has been judged scientifically suitable for publication and will be formally accepted for publication once it meets all outstanding technical requirements.

Within one week, you'll receive an e-mail detailing the required amendments. When these have been addressed, you'll receive a formal acceptance letter and your manuscript will be scheduled for publication.

An invoice for payment will follow shortly after the formal acceptance. To ensure an efficient process, please log into Editorial Manager at https://www.editorialmanager.com/pgph/ click the 'Update My Information' link at the top of the page, and double check that your user information is up-to-date. If you have any billing related questions, please contact our Author Billing department directly at authorbilling@plos.org.

Kind regards,

Biplab Kumar Datta, Ph.D.

Academic Editor